# The G protein alpha chaperone and guanine-nucleotide exchange factor RIC-8 regulates cilia morphogenesis in *Caenorhabditis elegans* sensory neurons

Christina M. Campagna, Hayley McMahon, Inna Nechipurenko *

Department of Biology and Biotechnology, Worcester Polytechnic Institute, Worcester, Massachusetts, United States of America

* inechipurenko@wpi.edu

**Data Availability Statement:** All relevant data are within the manuscript and its Supporting Information files.

## Abstract

Heterotrimeric G (αβγ) proteins are canonical transducers of G-protein-coupled receptor (GPCR) signaling and play critical roles in communication between cells and their environment. Many GPCRs and heterotrimeric G proteins localize to primary cilia and modulate cilia morphology via mechanisms that are not well understood. Here, we show that RIC-8, a cytosolic guanine nucleotide exchange factor (GEF) and chaperone for Gα protein subunits, shapes cilia membrane morphology in a subset of *Caenorhabditis elegans* sensory neurons. Consistent with its role in ciliogenesis, *C. elegans* RIC-8 localizes to cilia in different sensory neuron types. Using domain mutagenesis, we demonstrate that while the GEF function alone is not sufficient, both the GEF and Gα-interacting chaperone motifs of RIC-8 are required for its role in cilia morphogenesis. We identify ODR-3 as the RIC-8 Gα client and demonstrate that RIC-8 functions in the same genetic pathway with another component of the non-canonical G protein signaling AGS-3 to shape cilia morphology. Notably, despite defects in AWC cilia morphology, *ags-3* null mutants exhibit normal chemotaxis toward benzaldehyde unlike *odr-3* mutant animals. Collectively, our findings describe a novel function for the evolutionarily conserved protein RIC-8 and non-canonical RIC-8-AGS-3-ODR-3 signaling in cilia morphogenesis and uncouple Gα ODR-3 functions in ciliogenesis and olfaction.

## Author summary

Primary cilia are specialized cellular compartments that mediate communication between cells and their environment. In humans, deficits in cilia assembly and function manifest in genetic diseases called ciliopathies. While the molecular machinery of all major signaling pathways engaged in cell-cell and cell-environment communication is present inside cilia of different cell types, the mechanisms by which these signaling factors modulate cilia structure or cilia-dependent cellular functions are not well understood. In this study, we identify a new ciliary signaling module comprised of RIC-8 and Gα ODR-3 –key

**Funding:** This work was supported by the National Institute of Child Health and Human Development (R15 HD109706, IN https://www.nichd.nih.gov/) and Charles H. Hood Foundation Child Health Research Award (IN https://charleshoodfoundation.org). The funders had no role in study design, data collection and analysis, decision to publish, or preparation of the manuscript.

**Competing interests:** The authors have declared that no competing interests exist.

transducer of olfactory signaling–and find that RIC-8-dependent signaling is critically important for formation of specialized ciliary structures in *C. elegans* neurons. Our findings provide new insight into mechanisms of cilia assembly and highlight distinct functions of Gα ODR-3 in sensory transduction and cilia formation.

## Introduction

Primary cilia are hair-like cellular compartments that protrude from the surface of nearly all metazoan cell types [1]. Their core organization and assembly mechanisms are highly conserved across evolution. Cilia in different eukaryotic species concentrate molecular machinery of key signaling pathways, which in turn require intact cilia for proper transduction [2,3]. As a result, defects in cilia structure or function manifest in a spectrum of human genetic disorders called ciliopathies, many of which are associated with neurological deficits [4–7].

Heterotrimeric G (αβγ) proteins are canonical transducers of G protein coupled receptor (GPCR) signaling. Notably, many GPCRs and their downstream effectors localize to cilia in different cell types including neurons and can modulate cilia morphology [2,8–12]. For example, overexpression of the ciliary GPCR 5-HT6, which binds the neurotransmitter serotonin, induces the formation of excessively long cilia in cortical neurons [9]. In *Caenorhabditis elegans*, loss of ciliary Gα ODR-3 reduces cilia complexity in a subset of olfactory neurons [8,13]. In these cases, changes in cilia morphology are also accompanied by altered neuronal morphology and defective sensory behavior, respectively. Adenylate cyclase type III (ACIII) and the second messenger cyclic adenosine monophosphate (cAMP) at least in part mediate the effects of GPCR signaling on neuronal cilia morphology and function [9,14,15]. However, the regulators or downstream effectors of G protein signaling in the context of cilia morphogenesis or cilia-mediated neuronal functions remain unexplored.

In the canonical GPCR cascade, the ligand-activated receptor functions as a guanine-nucleotide exchange factor (GEF) that exchanges guanosine diphosphate (GDP) on the Gα subunit of the Gαβγ trimer for guanosine triphosphate (GTP), thus stimulating dissociation of Gα-GTP from Gβγ and allowing activation of their respective downstream targets [16]. Gα proteins are further classified into subfamilies (Gα$_s$, Gα$_{i/o}$, Gα$_q$, and Gα$_{12/13}$) based on sequence homology and effector specificity [17,18]. Gα signaling is terminated by the intrinsic GTPase activity of Gα and its subsequent reassociation with Gβγ.

In addition to the classical GPCR cascade, cytoplasmic factors can modulate the activity of heterotrimeric G proteins independently of receptors in non-canonical G protein signaling pathways (e.g. [19–21]). Resistance to inhibitors of cholinesterase-8 (RIC-8) is a highly conserved non-canonical GEF, which binds monomeric, GDP-bound Gα subunits and catalyzes the exchange of G-protein-bound GDP for GTP independently of GPCRs [22–24]. While there is only one *ric-8* gene in invertebrate genomes, mammals encode two RIC-8 homologs (RIC8A and RIC8B), which bind and activate distinct Gα clients [24,25]. *In vitro* biochemical studies further demonstrated that RIC-8 cannot activate Gα proteins found in the heterotrimeric complex with Gβγ [24]. However, *C. elegans*, *Drosophila*, and mammalian RIC-8 homologs can facilitate nucleotide exchange on GDP-bound Gα subunits that are sequestered by G Protein Regulatory (GPR/GoLoco) motif-containing proteins such as activator of G protein signaling 3 (AGS3) [26,27], LGN [28], Pins [29], and GPR-1/2 [30], thereby activating Gα in non-canonical G protein pathways.

In addition to their role as non-canonical Gα GEFs, RIC-8 proteins contribute to biogenesis of Gα across eukaryotic species. Specifically, RIC-8 functions as a chaperone that promotes

folding of nascent Gα subunits prior to their incorporation into the heterotrimeric complex with Gβγ and regulates their membrane localization [29,31–33]. RIC-8 has been implicated in regulating G protein signaling in many contexts ranging from sensory transduction [34,35] to asymmetric cell division [29,30,33,36–40] and synaptic transmission [41–43]. Despite ample *in vitro* evidence in support of GEF and chaperone activities of RIC-8 homologs, these functions have been challenging to uncouple *in vivo*.

Like in other animal cells, many GPCRs and Gα localize to cilia in *C. elegans* [8,13,44]. Unlike in mammals, where most cells are ciliated, only 60 sensory neurons form cilia in the adult *C. elegans* hermaphrodite. Twelve pairs of ciliated neurons reside in the bilateral sensory organs called amphids that are found in the worm head. While cilia in most amphid neurons display simple rod-like morphologies, AWA, AWB, and AWC amphid neurons possess cilia that are structurally complex and commonly referred to as 'wing' cilia [45–47]. Although cell-specific mechanisms contribute to cilia assembly and cargo trafficking in different sensory neuron types [44,48,49], the roles of G protein-dependent pathways in shaping cilia morphology are only starting to come into focus.

Here, we report a new role for RIC-8 in promoting cilia morphogenesis in a subset of *C. elegans* sensory neurons. Specifically, *ric-8* mutants exhibit defects in post-embryonic membrane expansion of specialized cilia in AWA, AWB, and AWC amphid sensory neurons. We further demonstrate that RIC-8 physically associates with Gα ODR-3 *in vivo* and positively regulates ODR-3 levels in AWC neurons consistent with its function as an ODR-3 chaperone. AGS-3 –a cytoplasmic GPR/GoLocco motif protein, which forms a transient tripartite complex with GDP-bound Gα and RIC-8, functions in the same genetic pathway with RIC-8 to regulate AWC cilia morphology. Interestingly, *ags-3* function appears to be dispensable for the AWC-mediated behavioral response to benzaldehyde. Collectively, our results identify a new non-canonical RIC-8-AGS-3-ODR-3 G protein signaling axis in morphogenesis of specialized cilia in sensory neurons and suggest that Gα ODR-3 plays distinct roles in AWC-mediated sensory behavior and ciliogenesis.

## Results

### RIC-8 is required for morphogenesis of specialized wing cilia during postembryonic development

The upstream regulatory sequences of the *ric-8* gene have been reported to contain a putative X-box–the binding motif for DAF-19 RFX-type transcription factor, which regulates expression of many canonical ciliary genes [50,51]. To determine if *ric-8* is required for ciliogenesis in sensory neurons, we first focused on morphologically complex 'wing' cilia that are present in the bilateral AWA, AWB, and AWC head sensory neurons. Each AWB neuron has two cilia of unequal length with membranous expansions in the distal ciliary regions (Fig 1A). Compared to wild type, the total length of AWB cilia was significantly reduced in three independent alleles of *ric-8*: *md303* and *md1909* hypomorphic and *ok98* null mutants (Fig 1A and 1B). Similarly, the maximal width of the distal AWB ciliary fans was significantly decreased in *ric-8 (md303)* and *ric-8(md1909)* mutant adults (Fig 1A and 1C and S3 Table), suggesting that *ric-8* regulates AWB cilia morphology. Since the majority of null *ric-8(ok98)* animals do not complete development and arrest prior to the L4 larval stage, we had to restrict cilia analysis to L3 larvae of this genotype. To determine if *ric-8* function in ciliogenesis extends to other neurons with wing cilia, we also examined AWA and AWC neurons. Cilia in AWA are extensively branched and exhibit a tree-like appearance, while AWC cilia have expansive membranous fans (Fig 1A). In *ric-8(ok98)* null mutants, the size of the AWA cilia was markedly reduced (Fig 1A and 1D). Similarly, the area of the AWC cilium was significantly smaller in the *ok98*

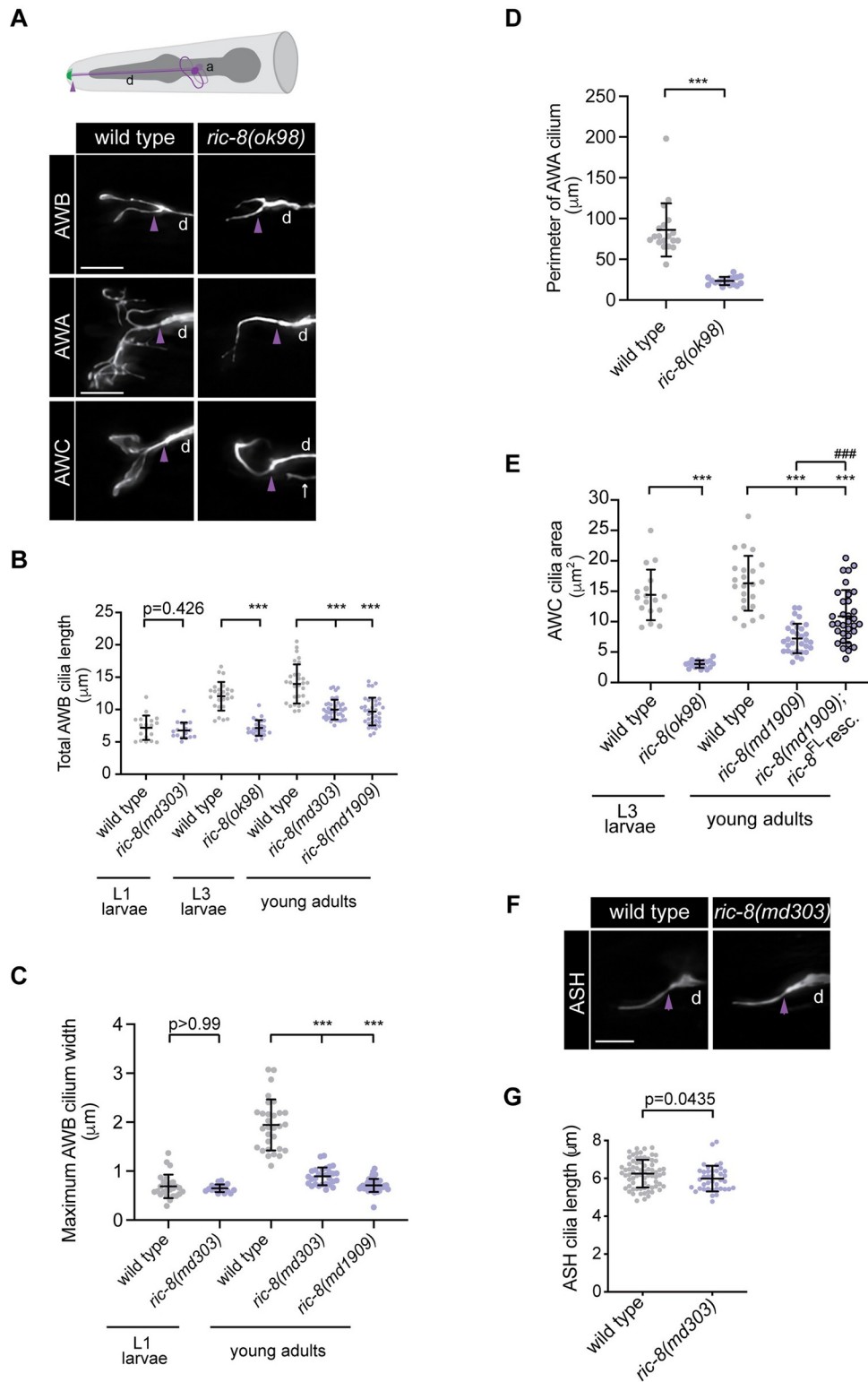

**Fig 1. RIC-8 is required for morphogenesis of specialized wing cilia.** (A) (Top) Cartoon of amphid (head) sensory neurons in *C. elegans*. Cilia are shown in green. (Bottom) Representative images of AWB, AWA, and AWC cilia in wild type and *ric-8(ok98)* null animals. AWB, AWA, and AWC neurons were visualized via expression of *str-1*p::GFP [52], *gpa-4*Δp::myr-GFP [53], and *ceh-36*p::GFP, which also labels ASE sensory neurons (white arrow) [54], respectively. (B–C) Quantification of total AWB cilia length (B) and total maximum width of AWB cilia (C) in animals of the

indicated genotypes and developmental stages. *** indicates different from wild type at p < 0.001 (Kruskal-Wallis with Dunn's multiple comparisons test). (D) Quantification of AWA cilium perimeter in wild type and *ric-8(ok98)* null L3 larvae. *** indicates different from wild type at p < 0.001 (Mann Whitney test). (E) Quantification of AWC cilium area in animals of the indicated genotypes and developmental stages. *** and ### indicates different from wild type and *ric-8 (md1909)*, respectively, at p < 0.001 (Brown-Forsythe and Welch ANOVA with Dunnett's T3 multiple comparisons test). (F–G) Representative images of ASH cilia (F) and quantification of ASH cilia length (G) in wild-type and *ric-8 (md303)* mutant adults. ASH neurons were visualized via expression of *osm-10*p::GFP. The p-value was calculated using Mann-Whitney test. In all image panels, anterior is at left; a–axon; d–dendrite; purple arrowheads mark cilia base; scale bars: 5 μm. In all scatter plots, means ± SD are indicated by horizontal and vertical black bars, respectively.

null and *md1909* hypomorphic alleles of *ric-8* compared to the age-matched wild-type controls (Fig 1A and 1E), indicating that *ric-8* regulates morphogenesis of all wing cilia types in *C. elegans*. The *ric-8*-dependent AWC cilia defects were partially but significantly rescued upon AWC-specific re-expression of full-length *ric-8* cDNA (isoform a) (Fig 1E and S3 Table). As detailed below, expression of the same *ric-8* isoform in all ciliated sensory neurons fully restored AWA cilia morphology. Together, the rescue experiments suggest that *ric-8* functions in ciliated neurons to regulate cilia morphogenesis. Since both hypomorphic alleles of *ric-8* exhibited qualitatively and quantitatively similar cilia defects to those observed in *ric-8(ok98)* null animals and were viable as adults, we focused all subsequent analyses on *md303* and/or *md1909* alleles.

Unlike AWA, AWB, and AWC neurons, most ciliated sensory neurons in *C. elegans* possess rod-like or 'channel' cilia at their dendritic tips [45,46]. To determine if *ric-8* also regulates morphogenesis of the structurally simple channel cilia, we next examined ASH neurons. We noted no gross differences in cilia morphology and only very mild, albeit statistically significant, difference in cilia length in *ric-8(md303)* strong reduction-of-function allele relative to wild type (Fig 1F and 1G and S3 Table), suggesting that RIC-8 is unlikely to play a major part in morphogenesis of channel cilia.

Ciliogenesis in most *C. elegans* neurons including the winged AWA, AWB, and AWC is initiated during late embryonic development, and axoneme elongation with accompanying membrane expansion continues post-embryonically [55,56]. To distinguish whether cilia defects observed in *ric-8* mutant wing neurons are due to impaired cilium initiation during embryogenesis or aberrant membrane expansion at larval stages, we quantified cilia length and maximal width of membranous fans in distal cilia of AWB neurons in newly hatched *ric-8 (md303)* L1 larvae. Strikingly, there was no difference in AWB cilia length or membranous fan width between wild type and *ric-8(md303)* mutant L1 larvae (Fig 1B and 1C and S3 Table), suggesting that *ric-8* functions during post-embryonic stages to promote expansion of ciliary membrane in winged neurons.

## RIC-8 localizes to cilia of *C. elegans* sensory neurons

RIC-8 has been previously reported to be expressed in ciliated sensory neurons [41,51,57]; however, its subcellular localization in ciliated cells has not been examined in detail. We expressed a functional (see Fig 1E) *tagrfp*-tagged *ric-8* cDNA (isoform a) under the *bbs-8* or *ceh-36Δ* promoters, which are transcriptionally active in all ciliated neurons or AWC, respectively [58], and observed RIC-8::TagRFP localization throughout ciliated sensory neurons including cilia. Notably, RIC-8::TagRFP localization in channel cilia was restricted to the proximal ~4-μm ciliary segment, with maximal fluorescence intensity around 1 μm distally from the cilia base (Fig 2A and 2B, see also S1 Fig and S3 Table).

Primary cilia in many cell types, including *C. elegans* neurons, contain a distinct proximal sub-compartment originally defined by localization of the INVS/NPHP-2 protein [61,62]. In

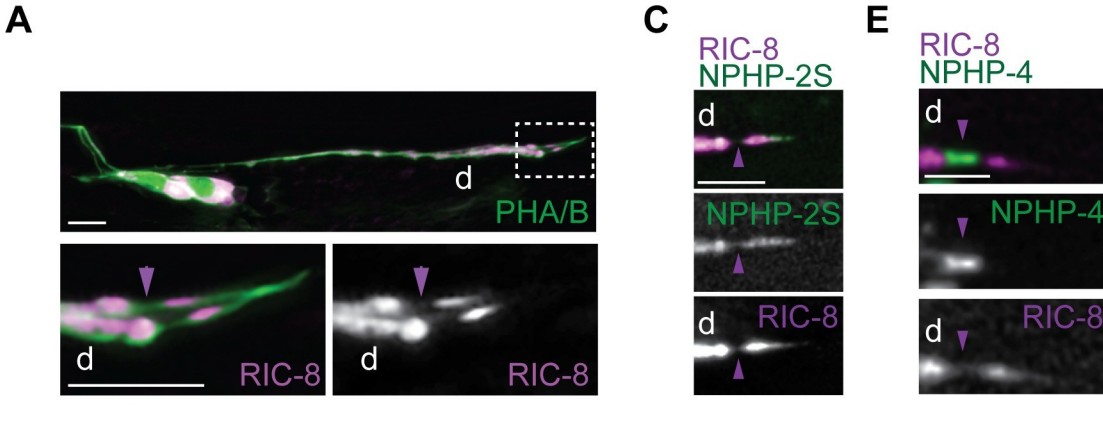

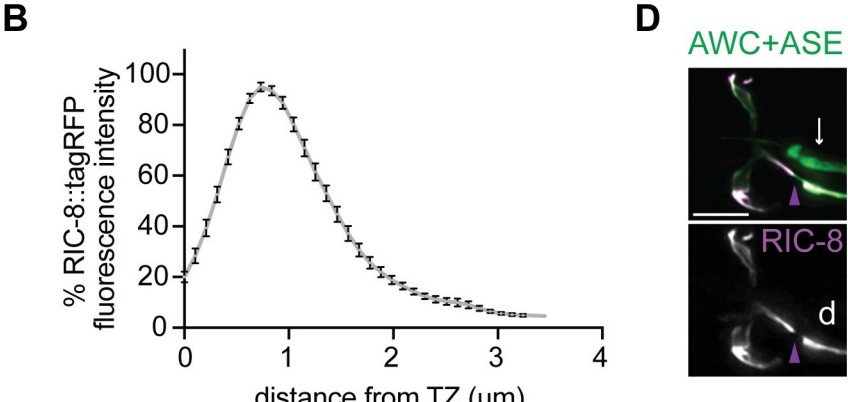

**Fig 2. RIC-8 localizes to cilia in sensory neurons.** (A) Representative images of RIC-8::TagRFP localization in phasmid neurons of wild-type adults. The phasmid neurons were visualized by expression of *bbs-8*p::myr-GFP. Phasmid cilia from the white dashed box (A, top) are magnified four-fold (A, bottom). (B) Line scan of RIC-8::TagRFP in phasmid cilia of wild-type adults. Zero on the x-axis represents the transition zone at cilia base. For each analyzed cilium, fluorescence intensities across the measured region were normalized to the maximum intensity for that cilium and expressed as the percent of this maximum intensity. Each point on the line scan represents the mean percent fluorescence intensity for a specific ciliary location across examined animals (n>20). Error bars are the SEM. (C) Images showing expression of RIC-8::TagRFP together with the inversin-compartment marker NPHP-2S::GFP [59] in phasmid neurons of wild-type adults. (D) Representative images of RIC-8::TagRFP localization in AWC neurons of wild-type adults. The AWC neurons were visualized by expression of *ceh-36*p::GFP, which also labels ASE sensory neurons (white arrow). (E) Representative images showing expression of RIC-8::TagRFP together with the transition-zone marker NPHP-4::GFP [60] in phasmid neurons of wild-type adults. In all image panels, anterior is at left; d–dendrite; purple arrowheads mark cilia base. Scale bars: 5 μm (A and D), 2.5 μm (C and E).

addition to INVS, several transmembrane signaling proteins localize to the inversin compartment of vertebrate and invertebrate sensory neurons, suggesting this region may play a role in sensory transduction [49,63,64]. We found that the localization pattern of RIC-8::TagRFP in channel cilia completely overlaps with that of NPHP-2S::GFP [59] (n = 20 PHA/B neurons) suggesting that RIC-8::TagRFP is enriched in the inversin compartment in this cilia type (Fig 2C). In contrast, RIC-8::TagRFP localized throughout the cilia in AWC neurons (Fig 2D). In both cilia types, RIC-8::TagRFP was excluded from the transition zone–a diffusion barrier that regulates entry and exit of molecular cargoes at the ciliary base and is demarcated by NPHP-4:: GFP (Fig 2D and 2E). Similar cell-specific differences in localization have also been demonstrated for other ciliary signaling molecules such as cyclic nucleotide gated channels [49] and

small G protein ARL-13 [65], suggesting that RIC-8 may contribute to regulation of the signaling modules that are restricted to the inversin compartment of channel cilia but distributed throughout the specialized wing cilia.

## The Gα chaperone activity of RIC-8 is required for cilia morphogenesis

Structural studies of the vertebrate RIC8A homolog revealed that this protein adopts a superhelical fold comprised of armadillo (ARM) and Huntington, elongation factor 3, PR65/A, TOR1 (HEAT) repeats [22,66]. RIC8A forms multiple contacts with its client Gα, and mutational analysis identified protein regions that are required for RIC8A bioactivity [22,67–69]. Specifically, the RIC8A fragment encompassing residues 1–453 (RIC8A$^{1-453}$) exhibited impaired GEF and chaperone activities; however, a smaller C-terminal truncation of RIC8A (RIC8A$^{1-492}$) displayed robust GEF activity but lost chaperone activity [27,69,70]. Since RIC-8 is highly conserved, we generated a RIC-8::TagRFP construct lacking the predicted chaperone and GEF domains (RIC-8$^{1-483}$) and a smaller C-terminal deletion construct that is predicted to have robust GEF activity but impaired chaperone activity (RIC-8$^{1-522}$) based on sequence alignment between the worm and mammalian RIC8A homologs [27,70] (Fig 3A). Like *ric-8 (ok98)* null animals, *ric-8(md303)* and *ric-8(md1909)* hypomorphs have smaller AWA cilia compared to wild type (Fig 3B and 3C and S3 Table). Expression of a full-length TagRFP-tagged RIC-8 construct (RIC-8$^{FL}$) in the *ric-8(md303)* background under the *bbs-8* regulatory sequences (*bbs-8*p) restored AWA cilia size to wild type (Fig 3B and 3C and S3 Table). On the other hand, expression of TagRFP-tagged RIC-8$^{1-483}$ construct in *ric-8(md303)* mutants or RIC-8$^{1-522}$ fragment in the *ric-8(md1909)* background failed to do so (Fig 3B and 3C and S3 Table), suggesting that the Gα chaperone activity is required and GEF activity alone is not sufficient for RIC-8 function in cilia morphogenesis.

Dual phosphorylation of rat RIC8A at serine 435 (Ser435) and threonine 440 (Thr440) by protein kinase CK2 is essential for efficient RIC8A binding to Gα subunits [23]. Mutation of the equivalent residues in *C. elegans* RIC-8 (Ser467 and Ser472) to alanines produced locomotor deficits resembling those of *ric-8* loss-of-function mutants [23]. To determine whether phosphorylation of *C. elegans* RIC-8 at Ser467 and Ser472 is similarly important for its function in ciliogenesis, we expressed the RIC-8::TagRFP construct with both residues mutated to alanines (RIC-8$^{S467A,Ser472A}$) in the *ric-8(md303)* background under the *bbs-8*p and found that this mutant construct also failed to rescue AWA cilia morphology (Fig 3B and 3C and S3 Table). Importantly, RIC-8::TagRFP mutant proteins localized normally to cilia and their overexpression in the wild-type background did not disrupt AWA cilia morphology (S1 Fig). These results indicate that lack of cilia morphology rescue upon expression of RIC-8$^{1-483}$ and RIC-8$^{S467A,Ser472A}$ constructs in *ric-8(md303)* mutants is not due to mislocalization or dominant-negative effects of the mutant RIC-8 protein variants. Together, these findings suggest that the chaperone activity and Gα binding of RIC-8 are necessary for cilia morphogenesis.

## Gα ODR-3 and RIC-8 function in the same genetic pathway and physically associate *in vivo* to regulate AWC cilia morphology

The *C. elegans* genome encodes 21 Gα subunits, the vast majority of which (17) are most similar to the mammalian Gα$_{i/o}$ class [71]. ODR-3, GPA-2, GPA-3, and GPA-13 Gα proteins have been previously shown to localize to AWC cilia [13,72]. Notably, only *odr-3* mutants have been reported to exhibit cilia defects in AWC [8,13]. To determine if ODR-3 is the Gα client of RIC-8 in AWC ciliogenesis, we first examined cilia morphology in AWC neurons of *odr-3 (n1605)* mutant animals. We found that cilia defects in *odr-3(n1605)* animals were qualitatively and quantitatively similar to those in *ric-8* mutants (Fig 4A and 4B and S3 Table). Additionally,

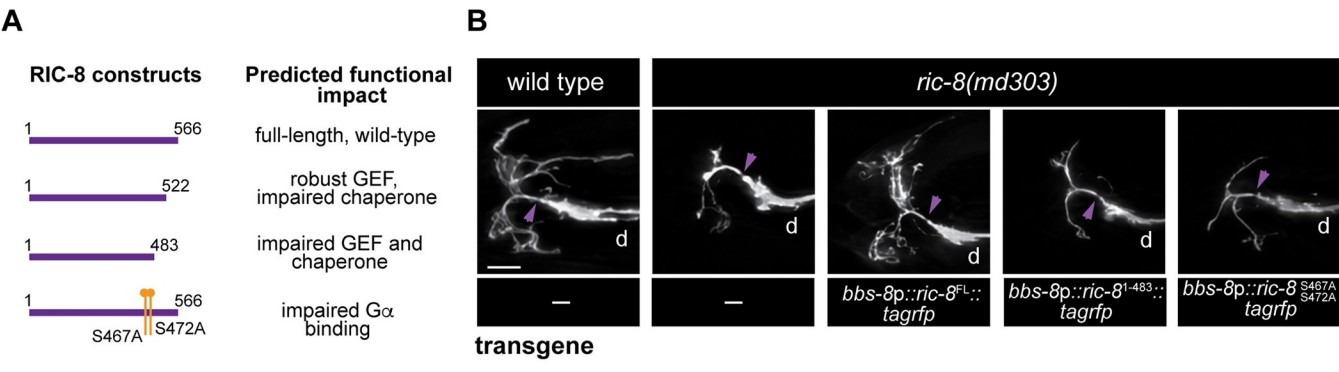

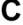

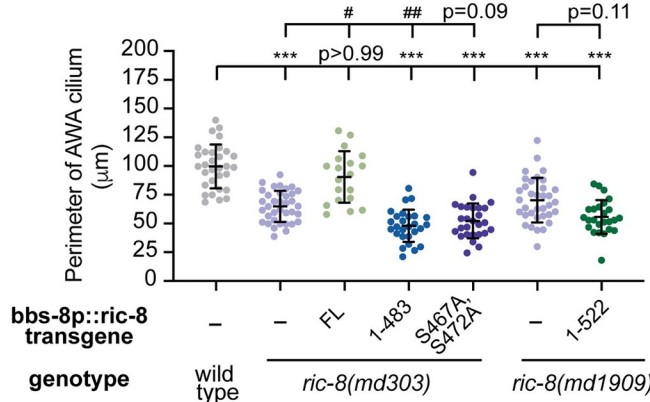

**Fig 3. The chaperone and Gα-interaction motifs of RIC-8 are necessary for cilia morphogenesis.** (A) RIC-8 mutant constructs and predicted functional outcomes of depicted mutations and truncations. (B–C) Representative images of AWA cilia (B) and quantification of AWA cilia perimeter (C) in adult animals of the indicated genotypes. In image panels, anterior is at left; d–dendrite; purple arrowheads mark cilia base. Scale bar: 5 μm. In the scatter plot, means ± SD are indicated by horizontal and vertical black bars, respectively. *** indicates different from wild type at p < 0.001; # and ## indicate different from *ric-8(md303)* at p < 0.05 and p < 0.01, respectively (Kruskal-Wallis with Dunn's multiple comparisons test).

AWC cilia in *ric-8(md1909); odr-3(n1605)* double mutants were indistinguishable from those in either single mutant (Fig 4B and S3 Table), suggesting that *ric-8* and *odr-3* function in the same genetic pathway to regulate cilia morphology.

As a complementary approach, we next investigated *in vivo* physical association between RIC-8 and Gα ODR-3 using bimolecular fluorescence complementation (BiFC) (Fig 4C) [74]. In BIFC, two non-fluorescent fragments of a fluorescent protein such as Venus will reconstitute an intact fluorescent protein (resulting in fluorescence) when fused to a pair of interacting protein partners. We chose BiFC over traditional biochemical approaches because both RIC-8 and ODR-3 are expressed in many cell types besides AWC [8,41], so BiFC would allow us to evaluate interaction between these proteins in the relevant *in vivo* context. Since formation of the bimolecular fluorescent complex can be irreversible [75], we expressed full-length RIC-8 tagged with the Venus C-terminal fragment (RIC-8$^{FL}$::VC) in wild-type animals under a heat-shock promoter (*hsp-16.2*p) [76] to ensure that results are not confounded by BiFC complex formation during development. Additionally, we expressed full-length ODR-3 tagged in an internal loop of its alpha-helical domain (see Materials and Methods) with the N-terminal Venus fragment (ODR-3$^{FL}$::VN) [77] in AWC neurons of the same animals. Importantly, we

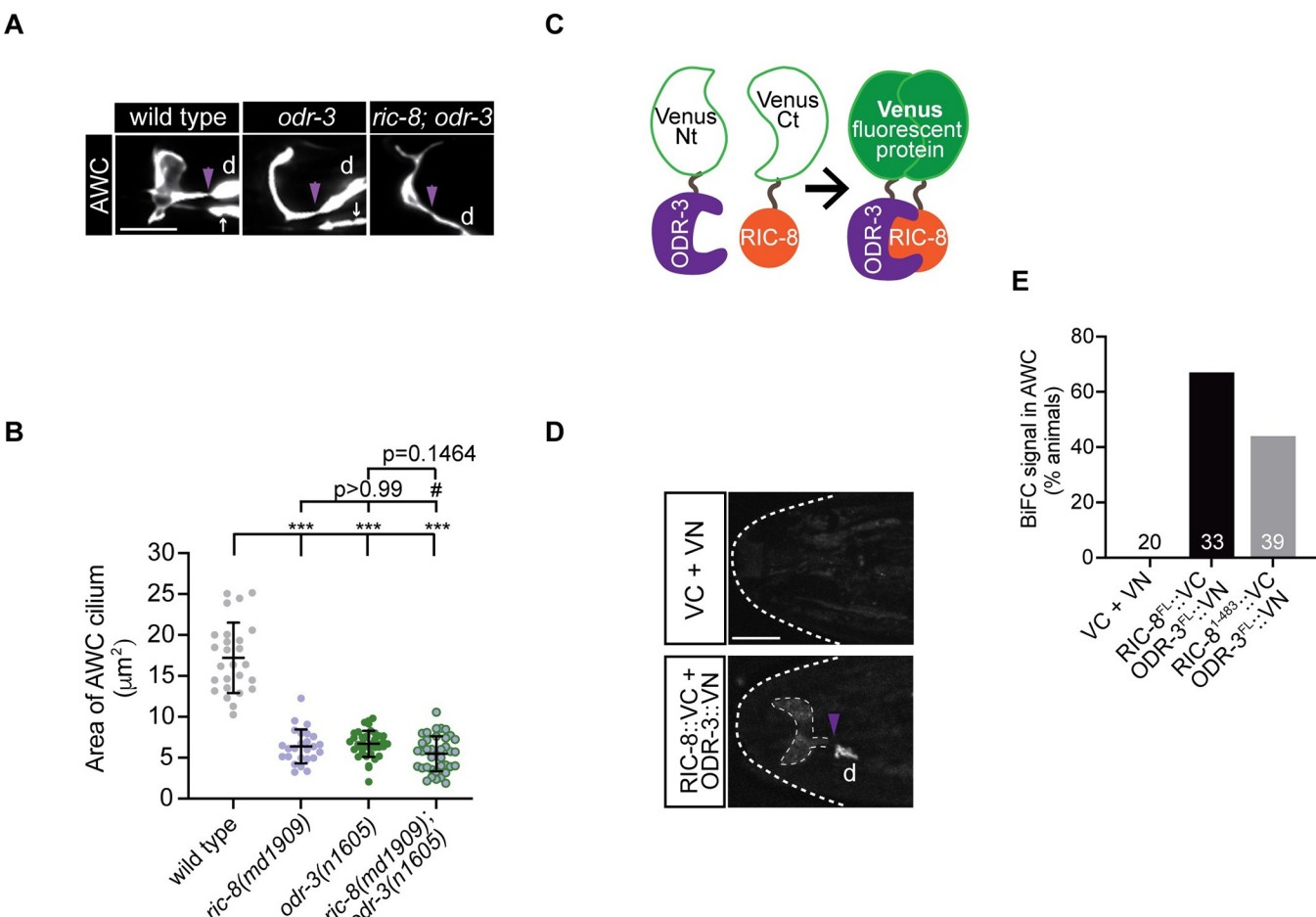

**Fig 4. RIC-8 and ODR-3 function in the same genetic pathway and associate *in vivo*.** (A) Representative images of AWC cilia in wild-type, *odr-3(n1605)*, and *ric-8(md1909); odr-3(n1605)* adult animals. The AWC neuron was visualized by expression of *ceh-36*p::GFP, which also labels ASE sensory neurons (white arrows). (B) Quantification of AWC cilium area in animals of the indicated genotypes. *** indicates different from wild type at p < 0.001 (one-way ANOVA with Dunnett's multiple comparisons test). # indicates different from *ric-8(md1909)* at p<0.05 (one-way ANOVA with Bonferroni's multiple comparisons test). Data for *ric-8(md1909)* are repeated from Fig 1E. Means ± SD are indicated by horizontal and vertical black bars, respectively. (C) Cartoon illustrating the basic principle of bimolecular fluorescence complementation (BiFC). (D) Representative images of the worm head in adult transgenic animals expressing indicated constructs. VN and VC:N- and C-terminal fragments of Venus, respectively. The heat-shock-inducible promoter *hsp-16.2*p [73] was used to drive expression of VC and RIC-8::VC; *ceh-36*Δp [54] was used to express VN and ODR-3::VN in AWC neurons. (E) Quantification of BiFC in the indicated transgenic animals. The number of analyzed animals per genotype is listed in the corresponding bars on the graph. In all image panels, anterior is at left; d–dendrite; purple arrowheads mark cilia base; scale bars: 5 μm.

demonstrate that the internal tag does not interfere with ODR-3 function, as the internally tagged ODR-3^FL::TagRFP fusion protein localizes to cilia when expressed in AWC (see Fig 5A) and rescues *odr-3*-dependent cilia defects (S2 Fig). We heat-shocked animals carrying *ric-8^FL::vc* and *odr-3^FL::vn* transgenes to induce expression of RIC-8^FL::VC in young adults and noted that 67% of heat-shocked transgenic animals exhibited reconstituted Venus inside AWC cilia and at cilia base (Fig 4D and 4E and S3 Table). Interestingly, we did not consistently observe BiFC along the entire length of the AWC dendrite. One potential explanation for this result is that ODR-3 levels outside the cilium are too low to be detected by BiFC, as ODR-3 is normally concentrated inside the AWC cilium [8] (see also Fig 5A). None of the heat-shocked transgenic animals carrying untagged N- and C-terminal Venus fragments expressed from the AWC-specific and heat-shock promoters, respectively, showed fluorescence complementation (Fig 4D and 4E).

Mammalian RIC8 proteins form several interaction interfaces with their client Gα proteins, including the C-terminus [67–69,78]. We next sought to determine if C-terminally truncated *C. elegans* RIC-8$^{1-483}$ that is unable to rescue *ric-8(md303)*-dependent cilia defects (see Fig 3) is still able to associate with ODR-3. Indeed, 44% of heat-shocked animals co-expressing *odr-3$^{FL}$*::*vn* and *ric-8$^{1-483}$*::*vc* exhibited Venus complementation (Fig 4E and S3 Table) suggesting that lack of rescue is not simply due to inability of truncated RIC-8 to bind Gα proteins. This result is consistent with findings in mammals, where an equivalent truncation of RIC8A appeared to weaken (but not fully abrogate) the interaction between RIC8A and its client Gα proteins [27,70]. Collectively, these results suggest that RIC-8 and ODR-3 associate *in vivo* in AWC, and that the C-terminus of RIC-8 may contribute to but is not essential for this interaction.

## RIC-8 positively regulates Gα ODR-3 levels in AWC neurons

RIC-8 depletion in several model organisms and mammalian cultured cells caused reduction in steady-state levels and plasma-membrane association of Gα proteins [29,32,33,37,39,79,80]. To determine whether RIC-8 regulates levels of Gα ODR-3 in AWC neurons, and thus functions as an ODR-3 chaperone, we examined localization of the internally tagged ODR-3$^{FL}$::TagRFP fusion protein in cilia, distal dendrites, and cell bodies of wild-type and *ric-8(md1909)* mutant animals. We confirmed that ODR-3::TagRFP is highly enriched inside the cilia of wild-type AWC neurons (Fig 5A), consistent with previously published findings [8]. We also observed low-level localization of tagged ODR-3 in the periciliary compartment (PCMC) of the distal dendrite and cell bodies (Fig 5B and S3 Table). While ODR-3::TagRFP similarly localizes to cilia of *ric-8(md1909)* mutant AWC, the overall levels of the fusion protein are markedly reduced throughout the neuron, including the cilium (Fig 5A and 5B and S3 Table). Furthermore, ODR-3::TagRFP localization inside cilia of *ric-8(md1909)* mutants appears more punctate compared to wild type (compare top and bottom panels in Fig 5A), suggesting that RIC-8 may also facilitate ODR-3 association with the ciliary membrane. Notably, RIC-8::GFP was still present in AWC cilia of all examined *odr-3(n1605)* mutants (n = 30) (Fig 5C),

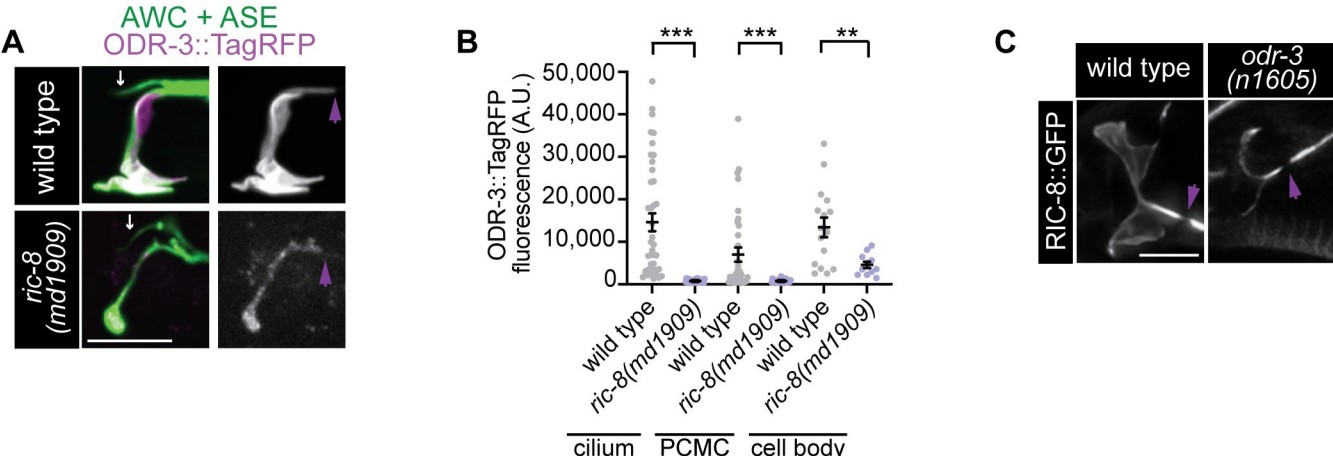

**Fig 5. RIC-8 positively regulates ODR-3 levels in AWC neurons.** (A) Representative images of ODR-3::TagRFP localization in wild-type and *ric-8(md1909)* mutant adults. AWC was visualized via expression of *ceh-36*p::GFP, which also labels ASE sensory neurons (white arrows). (B) Quantification of ODR-3::TagRFP fluorescence intensity in the indicated cellular compartments of wild-type and *ric-8(md1909)* mutant adults. PCMC–periciliary membrane compartment. Means ± SD are indicated by horizontal and vertical black bars, respectively. ** and *** indicate different from wild type at p < 0.01 and p < 0.001, respectively (Mann Whitney test). (C) Representative images of RIC-8::GFP localization in wild-type and *odr-3(n1605)* mutant adults. In all image panels, anterior is at left; purple arrowheads mark cilia base; scale bars: 5 μm.

indicating that RIC-8 localization to AWC cilia does not require Gα ODR-3. Taken together, these findings show that RIC-8 positively regulates ODR-3 protein abundance in AWC neurons and are consistent with RIC-8 acting as a Gα ODR-3 chaperone.

## AGS-3 is required for AWC cilia morphogenesis

In non-canonical (i.e. receptor-independent) G protein signaling, heterotrimeric G proteins are activated by cytoplasmic GEFs such as RIC-8 and can regulate downstream effectors that are distinct from those in the canonical GPCR cascade [81–84]. Previous studies have demonstrated that mammalian RIC8A can activate Gα subunits that are bound to the cytoplasmic GPR/GoLoco-motif protein AGS3 [26,27] as part of non-canonical G protein signaling. Like RIC-8, *C. elegans* AGS-3 is broadly expressed in the nervous system including sensory neurons in the worm head and has been shown to function cooperatively with RIC-8 to modulate food-seeking behavior [85]. Therefore, we next hypothesized that RIC-8 functions together with AGS-3 to shape cilia morphology. To test this hypothesis, we first examined AWC cilia morphology in *ags-3(ok1169)* null mutants [85]. We found that the area of AWC cilia was significantly reduced in these mutant animals, and their overall cilia morphology appeared remarkably similar to that of *ric-8* mutants (Fig 6A and 6B and S3 Table), indicating that *ags-3* is required for cilia morphogenesis in AWC. Moreover, *ric-8(md1909); ags-3(ok1169)* double mutants were statistically indistinguishable from *ric-8(md1909)* and *ags-3(ok1169)* single mutants (Fig 6B and S3 Table), suggesting that *ric-8* and *ags-3* function in the same genetic pathway to regulate cilia morphology.

We next wanted to determine whether the ciliary levels of ODR-3 were altered in the absence of *ags-3* function, similarly to *ric-8* mutants. To that end, we expressed full-length ODR-3::TagRFP in AWC neurons of *ags-3(ok1169)* mutants and noted that ODR-3::TagRFP localized normally to AWC cilia and appeared to be membrane-associated (Fig 6C and 6D and S3 Table). This result suggests that *ags-3* loss does not affect the chaperone activity of RIC-8 toward ODR-3. Interestingly, overexpression of a full-length, wild-type ODR-3::TagRFP in the *ags-3* null background restored the area of AWC cilia to nearly wild type (Fig 6B and 6C and S3 Table). The rescue was dependent on intact RIC-8, as *ric-8(md1909); ags-3(ok1169)* double mutants overexpressing wild-type ODR-3::TagRFP displayed the same AWC cilia defects as those observed in *ric-8(md1909); ags-3(ok1169)* animals without the *odr-3::tagrfp* transgene (Fig 6B and S3 Table). Collectively, these findings are in line with the model that *C. elegans* AGS-3 binds the inactive GDP-bound ODR-3 in AWC neurons, thereby preventing its re-association with Gβγ and allowing RIC-8-catalyzed nucleotide exchange to activate ODR-3. The RIC-8-dependent activation of ODR-3 in turn promotes expansion of ciliary membrane in AWC neurons. In this scenario, loss of *ags-3* would prompt rapid re-association of Gα with Gβγ thereby limiting the ability of RIC-8 to activate ODR-3 in the non-canonical pathway and would lead to cilia morphogenesis defects. On the other hand, ODR-3 overexpression in the *ags-3* mutant background likely generates excess of free ODR-3, which is not bound by Gβγ, and thus available for binding and activation by RIC-8.

## AGS-3 is not required for canonical GPCR-ODR-3-dependent olfactory behavior

We next wanted to test if AGS-3 is required for AWC-mediated olfactory behavior, which is mediated by canonical GPCR-ODR-3 signaling [86]. To this end, we assessed the ability of *ags-3(ok1169)* animals to chemotax toward a volatile odorant benzaldehyde, which is sensed by the AWC neurons [87]. In population chemotaxis assays, wild-type adult hermaphrodites exhibit robust attraction to a range of benzaldehyde concentrations, while *odr-3(n1605)*

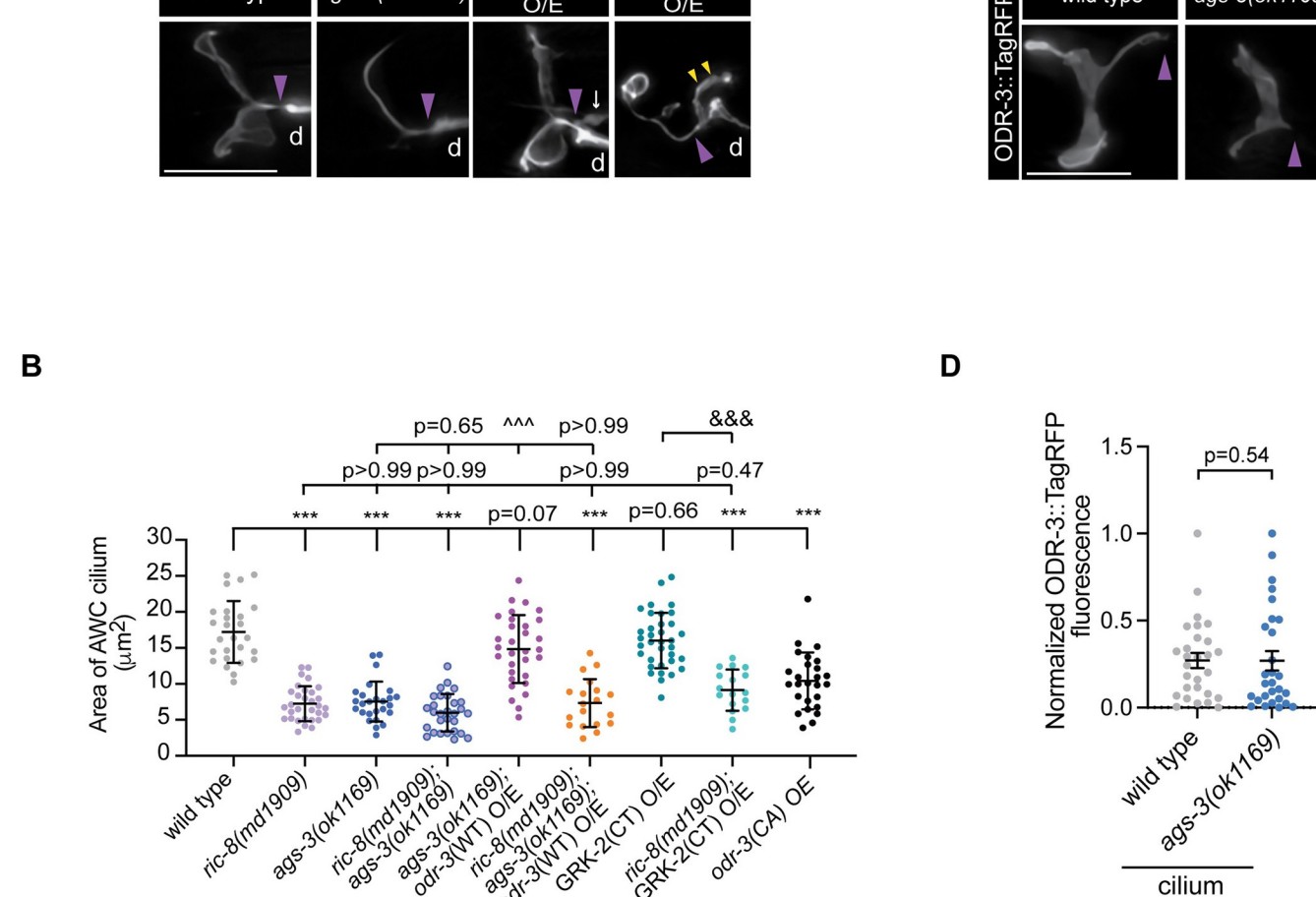

**Fig 6. AGS-3 is required for AWC cilia morphogenesis.** (A–B) Representative images (A) and quantification (B) of the AWC cilium area in adult animals of the indicated genotypes. AWC was visualized via expression of *ceh-36*p::GFP, which also labels ASE sensory neurons (white arrow). Data for wild type and *ric-8 (md1909)* are repeated from Fig 4B. Means ± SD are indicated by horizontal and vertical black bars, respectively. *** indicates different from wild type at $p < 0.001$. ^^^ and &&& indicate different from *ags-3(ok1169)* and GRK-2^CT O/E, respectively, at $p < 0.001$ (One-way ANOVA with Bonferroni's multiple comparisons test). (C–D) Representative images (C) and quantification (D) of ODR-3::TagRFP localization in AWC cilia of wild-type and *ags-3(ok1169)* mutant adults. Means ± SEM are indicated by horizontal and vertical black bars, respectively. The p-value was calculated using Mann-Whitney test. In all image panels, anterior is at left; d–dendrite; purple arrowheads mark cilia base; scale bars: 5 μm.

mutants show a markedly reduced response to this odorant [8,13,87] (S3 Fig and S3 Table). Strikingly, *ags-3(ok1169)* null hermaphrodites exhibited strong attraction to benzaldehyde (1:200), similar to wild-type animals (S3 Fig and S3 Table). This outcome suggests that while the non-canonical AGS-3-G-protein signaling is required for cilia morphogenesis, it is largely dispensable for response to benzaldehyde.

## Gβγ sequestration does not alter AWC cilia morphology

Since AGS3 stabilizes GDP-bound inactive $G\alpha_i$ subunits and prevents their re-association with Gβγ and GPCRs [84,88–90], AGS3 may augment signaling through Gβγ [91,92]. To assess the role of Gβγ signaling in cilia morphogenesis, we overexpressed the carboxyl terminus of GRK-2 (GRK-2^CT O/E), which binds to and blocks Gβγ interaction with the effectors

[93,94], in AWC neurons of wild-type animals. Blocking Gβγ signaling by GRK-2CT overex-pression in the wild-type background had no effect on AWC cilia morphology (Fig 6A and 6B and S3 Table), suggesting that Gβγ signaling is unlikely to play a critical role in this process. Similarly, overexpression of GRK-2CT in *ric-8(md1909)* mutants that have low levels of Gα ODR-3 (see Fig 5) was not sufficient to rescue cilia morphology (Fig 6B and S3 Table), indicat-ing that cilia defects in *ric-8* mutants are unlikely to be caused by excess of free Gβγ subunits. In contrast, overexpression of constitutively activated ODR-3$^{Q206L}$ (ODR-3$^{CA}$ O/E) in wild-type animals caused reduction in the area of AWC cilia in agreement with the earlier studies [8] (Fig 6A and 6B and S3 Table). Interestingly, AWC distal dendrites in ODR-3$^{CA}$ O/E mutant animals also exhibited marked membranous extensions at the cilia base (wild type: 14% of AWC neurons with dendritic extensions, n = 29; *odr-3*$^{CA}$ O/E: 100% of AWC had den-dritic extensions, n = 24). These ectopic membrane protrusions are reminiscent of the extra-cellular vesicles that have been reported to bud from the cilia base of sensory neurons [95–97]. Collectively, our findings are consistent with the hypothesis that Gα ODR-3 signaling, rather than βγ, mediates AWC cilia morphogenesis.

## Discussion

Here we report a new cellular function for RIC-8 –a highly conserved non-canonical GEF and chaperone for Gα proteins. Loss of *ric-8* function leads to dramatic changes in complex cilia morphologies of wing neurons. Specifically, we show that *ric-8* regulates post-embryonic expansion of ciliary membrane in these neuron types by functioning as a GEF and chaperone for Gα ODR-3. We also find that another component of the non-canonical G protein signaling AGS-3 is similarly required for AWC cilia morphogenesis–a novel function for this highly conserved protein. Since our data show that βγ signaling is likely dispensable for cilia morpho-genesis, we propose that AGS-3 shapes cilia morphology by facilitating RIC-8-mediated ODR-3 activation, in a manner similar to its mammalian homologs [26,27]. Specifically, AGS-3 likely stabilizes GDP-bound Gα ODR-3 and prevents its reassociation with Gβγ subunits, thus allow-ing RIC-8 to bind and activate ODR-3 in the non-canonical receptor-independent pathway (Fig 7).

Our results also suggest that Gα ODR-3 has at least two distinct functions in AWC neu-rons–it functions in the canonical GPCR cascade to mediate olfactory behavior and in the non-canonical signaling pathway with RIC-8 and AGS-3 to promote cilia membrane biogene-sis (Fig 7). Consistent with this model, *ags-3* null mutants exhibit normal chemotaxis toward benzaldehyde despite having severe AWC cilia defects. Uncoupling of some sensory signaling aspects from cilia morphology in *C. elegans* wing neurons has been proposed before [8,13,98]; however, our work provides a first glimpse into the potential mechanistic basis for this phe-nomenon. It has been previously demonstrated that AWA neurons in mutants with severely perturbed cilia morphology showed robust responses to diacetyl–a volatile odorant detected by AWA [98]. However, the same mutants were defective in AWA desensitization and habitu-ation [98], suggesting that cilia play a nuanced role in shaping sensory responses in these neu-rons. Similarly, the outer segments of cone photoreceptors in the *peripherin* mutant mouse are severely disorganized, yet they remain capable of phototransduction with only mildly reduced sensitivity [99]. It would be interesting to explore the extent to which AWC cilia morphology and/or non-canonical AGS-3-mediated signaling contribute to olfactory sensitivity, habitua-tion to AWC-sensed odorants [100], and plasticity [101]. These studies would provide an important insight into the functions of specialized cilia morphologies in sensory biology.

Is RIC-8 required for AWC-mediated chemotaxis? RIC-8 functions as a Gα chaperone, and ODR-3 levels are markedly reduced in AWC neurons of *ric-8* mutants. Therefore, AWC-

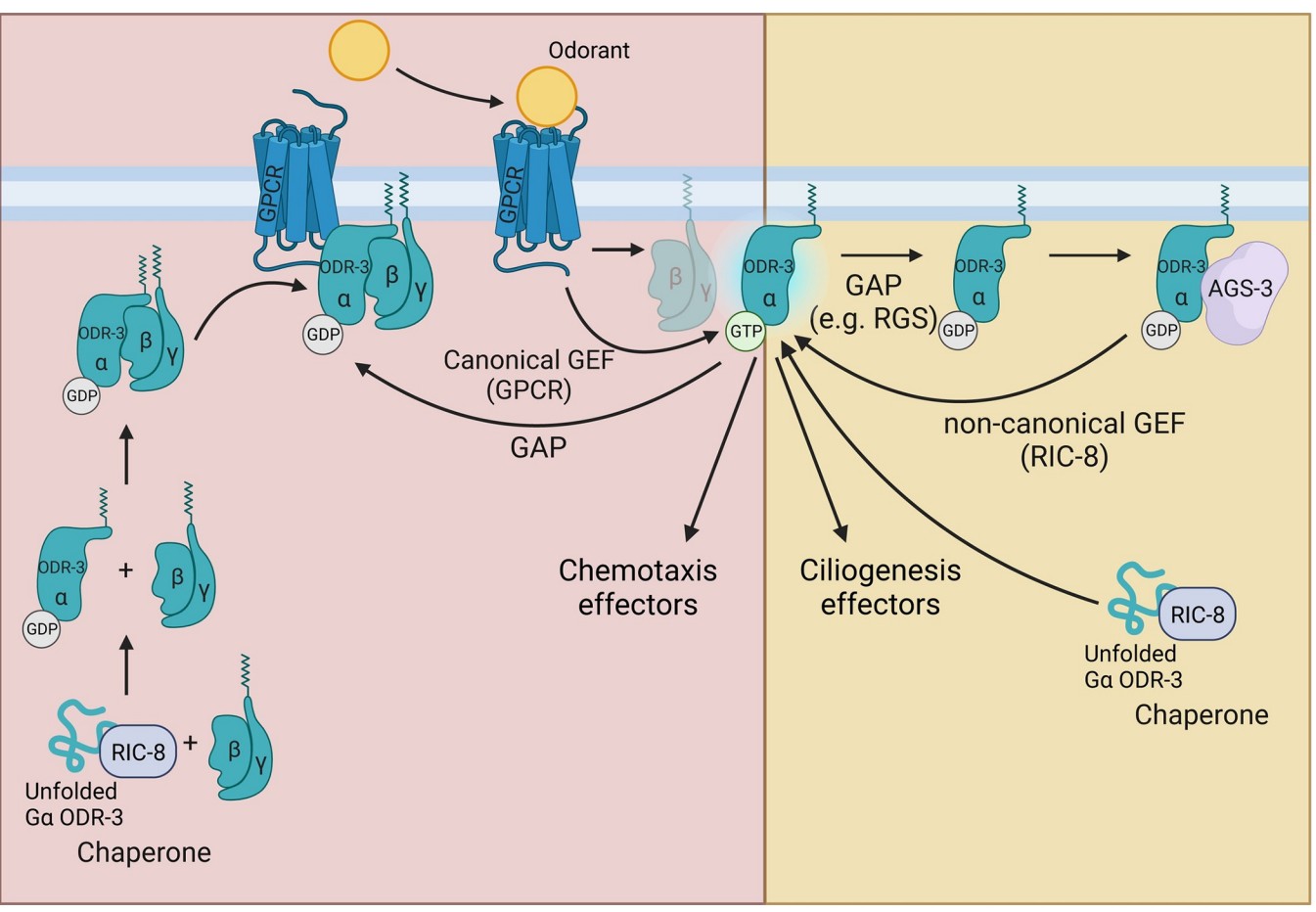

**Fig 7. Proposed model: non-canonical RIC-8-AGS-3-ODR-3 signaling shapes cilia morphology in AWC neurons.** RIC-8 functions as a chaperone to promote folding of nascent Gα ODR-3, which is ultimately incorporated in a heterotrimeric complex with Gβγ and associates with a GPCR. (Left) ODR-3 functions in the canonical GPCR cascade to mediate olfactory behaviors (e.g. attraction to benzaldehyde). Odorant-bound GPCR functions as a GEF to activate Gα ODR-3, which in turn disengages from Gβγ and transduces olfactory signals to the downstream effectors (e.g. GCs and CNG channels) [86]. Upon GTP hydrolysis (GAP), GDP-bound ODR-3 reassociates with Gβγ and GPCR. (Right) During cilia morphogenesis, ODR-3 is activated by a non-canonical GEF RIC-8 independently of GPCRs. AGS-3 binds the GDP-bound ODR-3, thereby preventing it from reassociation with Gβγ, and facilitates RIC-8-mediated exchange of GDP for GTP resulting in ODR-3 activation. This figure was created with BioRender.com.

mediated olfactory behaviors that require ODR-3 (e.g. attraction to benzaldehyde) will be likely impaired in the absence of *ric-8* due to low levels of ODR-3 in this genetic background. Since global knockout of *ric-8* results in full-body paralysis due to the role of *ric-8* in regulating neurotransmitter release at the neuromuscular junctions [41], we were unable to assess olfactory behaviors in this genetic background. In addition to AWC, both RIC-8 and ODR-3 are co-expressed in several ciliated sensory neurons including AWA, AWB, and ASH [8]. Furthermore, *odr-3* mutants exhibit defects in AWA-, AWB- and ASH-dependent olfactory behaviors [8,87,102]. Therefore, it would be interesting to determine experimentally whether *ric-8* regulates odor responses and/or olfactory plasticity in AWC and/or contributes to olfactory signaling in AWA, AWB, and ASH sensory neurons by generating cell-specific *ric-8* knockout animals or measuring calcium dynamics in these neurons following relevant sensory stimuli in global *ric-8* knockouts.

How might RIC-8-ODR-3 signaling regulate cilia morphogenesis? In cultured mammalian cells, RIC8A signaling modulates spindle positioning by controlling localization of cytoplasmic dynein to cell cortex [40]. In addition to its role in spindle organization, cytoplasmic dynein

drives microtubule-based vesicular transport of ciliary cargoes, including rhodopsin, to the cilia base [103–105]. Interestingly, zebrafish Dynein1 is required for morphogenesis of the photoreceptor outer segment, which represents a uniquely modified primary cilium, as well as post-Golgi vesicle trafficking [106]. Therefore, it would be an exciting future direction to explore the effects of RIC-8-Gα signaling on dynein-mediated trafficking of ciliary membrane components, particularly during the larval stages, when wing cilia undergo marked membrane expansion.

Mammalian RIC-8 homologs have been reported to potentiate G protein signaling in several canonical GPCR pathways. For example, RIC8B enhanced cAMP production in response to odorant receptor activation [35], while RIC8A amplified $G\alpha_q$- and $G\alpha_{12}$-mediated transduction following stimulation of muscarinic acetylcholine receptor and hTas2R16 taste receptor, respectively [34,107]. In the canonical olfactory pathway, the Gα ODR-3 is posited to function downstream from GPCRs [8] to inhibit guanylyl cyclases that produce cGMP [86]. Cyclic nucleotide-dependent pathways have been demonstrated to modulate cilia length in mammals and contribute to remodeling of complex cilia morphologies in *C. elegans* neurons [108,109]. Therefore, one intriguing possibility is that *C. elegans* RIC-8 shapes morphology of specialized wing cilia at least in part by modulating cGMP signaling.

## Materials and methods

### *C. elegans* genetics

*C. elegans* strains were maintained at 20˚C on standard nematode growth media (NGM) plates seeded with the OP50 strain of *Escherichia coli*. *C. elegans* strain N2 (variety Bristol) was used as wild type. Standard genetic techniques were used to create double-mutant strains and to introduce transgenes into mutant backgrounds. All mutant genotypes were confirmed by PCR and/or Sanger sequencing. Transgenic *C. elegans* were generated by standard germline transformation. To generate transgenic animals, all plasmids were injected at 5–40 ng/µL together with the *unc-122*Δp::*gfp* or *unc-122*Δp::*dsred* co-injection markers (injected at 30 and 40 ng/µL, respectively). At least two independent lines were examined for each transgene, and the same transgenic array was examined in wild-type and corresponding mutant backgrounds.

A complete list of strains used in this manuscript is provided in S1 Table.

### Molecular biology

*cDNA constructs*: AWC-specific and pan-ciliary expression of transgenes was achieved by subcloning relevant cDNAs downstream from 0.7-kb of *ceh-36* (*ceh-36*Δp) [54] and ~0.9-kb of *bbs-8* regulatory sequences [58], respectively, into a modified pPD95.77 (a gift from A. Fire) or pMC10 (a gift from M. Colosimo) *C. elegans* expression vectors [58]. Coding sequences of *ric-8* and *nphp-2s* were amplified from a mixed-stage N2 cDNA library using Phusion high-fidelity DNA polymerase (NEB) with gene-specific primers and verified by Sanger sequencing.

*ric-8 mutant constructs*: Functional RIC-8 domains were identified based on homology with human and rat RIC8A. Truncated *ric-8* constructs were generated by PCR using high-fidelity Phusion DNA polymerase (NEB). Phosphorylation mutants were created using the QuikChange Lightning site-directed mutagenesis kit (Agilent Technologies). All mutant constructs were verified by Sanger sequencing.

*TagRFP-tagged odr-3 constructs*: *odr-3* coding sequences corresponding to amino acids (AA) 1–118 and AA 119–356 were amplified from a mixed-stage N2 cDNA library using Phusion DNA polymerase (NEB) with gene-specific primers. *Tagrfp* cDNA with flanking DNA segments encoding SGGGGS linkers was amplified using Phusion DNA polymerase (NEB) and inserted between Gly118- and Glu119-encoding codons of *odr-3* [77] in the modified

pMC10 vector using NEBuilder HiFi DNA assembly (NEB). The resulting plasmid sequence was confirmed by Sanger sequencing. The same plasmid was subjected to site-directed mutagenesis using QuikChange Lightning kit (Agilent Technologies) to generate the TagRFP-tagged constitutively active *odr-3* mutant (ODR-3$^{Q206L}$).

*BiFC constructs*: cDNA sequences encoding N- and C-terminal fragments of Venus were amplified from pCe-BiFC-VN173 (Addgene) and pCe-BiFC-VC155 (Addgene) plasmids, respectively. The cDNA fragments encoding N- and C-termini of Venus were subsequently subcloned into modified *C. elegans* pMC10 expression vectors containing *ceh-36*Δp::*odr-3* and *hsp-16.2*p::*ric-8* sequences, respectively, using NEBuilder HiFi assembly or standard restriction enzyme cloning. The N-terminal Venus fragment was flanked with SGGGGS linkers on 5' and 3' ends and inserted between Gly118 and Glu119 of ODR-3. The plasmids were verified by Sanger sequencing.

*C-terminal GRK-2 fragment (GRK-2$^{CT}$)*: the C-terminal βγ-binding domain of *C. elegans* GRK-2 was identified based on homology with human GRK2. *grk-2* cDNA corresponding to the C-terminal GRK-2 fragment was amplified from a mixed-stage N2 cDNA library using Q5 high-fidelity DNA polymerase (NEB) with gene-specific primers, subcloned into pMC10 *C. elegans* expression vector downstream from the *ceh-36*Δp using standard cloning techniques, and confirmed by Sanger sequencing.

A complete list of DNA constructs used to generate transgenic strains in this work is provided in S2 Table.

## Microscopy

L1 and L3 hermaphrodite larvae and one-day-old adults were immobilized in 10mM tetramisole hydrochloride (MP Biomedicals) and mounted on 10% agarose pads placed on microscope slides. Animals were imaged on an upright THUNDER Imager 3D Tissue (Leica). Complete *z*-stacks of ciliated neurons (ASH, AWA, AWB, and AWC) were acquired at 0.22μm intervals with K5 sCMOS camera (Leica) in Leica Application Suite X software using an HC Plan Apochromat 63X NA 1.4–0.60 oil immersion objective. For BiFC and experiments examining ODR-3::TagRFP levels in wild-type and *ric-8(md1909)* mutant AWC neurons (Fig 5), adult animals were imaged on an inverted Nikon Ti-E microscope with Yokogawa CSU-X1 spinning disk confocal head using 60X NA 1.40 oil immersion objective. Complete *z*-stacks of either AWC cilia or entire AWC neurons were acquired in MetaMorph 7 (Molecular Devices).

## Bimolecular fluorescence complementation assay

Transgenic worms expressing complementary pairs of BiFC constructs were picked as L4 larvae and allowed to age to one-day-old adults at 20˚C overnight on NGM plates seeded with OP50. To induce expression of the full-length *ric-8* cDNA fused to the C-terminal Venus fragment (RIC-8::VC) or the VC fragment alone under the *hsp-16.2*p promoter, young adult hermaphrodites were heat-shocked at 33˚C for two hours and allowed to recover at 20˚C for 1–2 hours prior to imaging. Following recovery, animals were imaged as described in the Microscopy section above. For each genotype, BiFC was quantified from images acquired on at least two independent days.

## Chemotaxis assay

Chemotaxis assays were performed essentially as previously described [87]. Briefly, assay plates were prepared the day of the assay by adding 10 mL of 2% Difco-agar (BD) in assay buffer (5 mM potassium phosphate, pH 6.0, 1 mM CaCl$_2$, and 1 mM MgSO$_4$) to 10-cm round petri dishes. After agar was solidified, 1 μL of 1M sodium azide was placed at opposite ends of the

assay plate and allowed to soak in. Young adult hermaphrodites were washed off their growth plates with cholesterol-free S Basal into 1.5-mL microcentrifuge tubes and allowed to settle to the bottom. The wash step was repeated twice with S-Basal before animals were resuspended in Milli-Q water, and approximately 200 worms were placed at the origin, which was equidistant from two sodium azide spots. One microliter of the odorant (benzaldehyde, Sigma) dissolved in ethanol (200 proof, Sigma) or solvent control (ethanol) were added to the sodium azide spots, the petri-dish lid was closed, and animals within a 2-cm radius of the odorant and solvent spots were counted 60 minutes later. Chemotaxis index (CI) was calculated as follows:

CI = (# worms at attractant—# worms at ethanol)/ total worms that left origin

Assays were performed in triplicate with all genotypes tested in parallel and repeated on at least three separate days.

## Image analysis

Analysis of fluorescence microscopy images was carried out using Fiji/Image J (National Institute of Health, Bethesda, MD). Cilia morphology and fusion protein localization were quantified from images collected on at least two independent days. For experiments examining localization and fluorescence intensity of fusion proteins, all genotypes were grown in parallel and imaged at identical settings. Analysis-specific methods are detailed below.

*Cilia length (ASH, AWB)*: Line segments from cilia base to cilia tip were drawn using straight- or segmented-line tools from maximum intensity projections generated in Fiji/Image J. For AWB neurons, lengths of both cilia were summed for each neuron.

*Maximum cilia width (AWB)*: Straight-line segments were drawn across the widest point of each AWB cilium from maximum intensity projections generated in Fiji/Image J. Width measurements from two AWB cilia of each neuron were summed and plotted.

*Cilia area (AWC)*: The *z* slices that encompassed the AWC cilium in its entirety (lateral view) were rendered into maximum- or average-intensity projections in Fiji/Image J. AWC cilia were outlined using freehand selection tool, and the area enclosed by the region of interest (ROI) was measured and plotted. To improve clarity of the projected images and accuracy of measurements, in some cases, different regions of the ciliary fan were projected and measured separately, and measurements from all regions of the same cilium were subsequently summed.

*Cilia perimeter (AWA)*: Only images that contained two AWA neurons in clearly separate planes were used for analysis to ensure that only one cilium was measured per animal. The z-slices that encompassed one AWA cilium in its entirety were rendered into maximum-intensity projections in Fiji/Image J. The ROI was defined by outlining AWA cilia with the freehand selection tool, and the image outside of the ROI was cleared using the "clear outside" command. The ROI thresholding was adjusted to ensure that the thresholded region matched cilia morphology, and the perimeter of the thresholded region (i.e. AWA cilium) was measured and plotted.

*Length of ciliary RIC-8::TagRFP signal (PHA/B)*: To measure the length of RIC-8::TagRFP enrichment in channel cilia, *z*-slices containing cilia of PHA/B neurons were rendered into maximum-intensity projections. Straight-line segments were drawn from base to tip of the ciliary RIC-8::TagRFP signal in non-overlapping PHA/B neurons.

*Line scans (PHA/B)*: Line scans were generated by drawing a straight line from cilia base to the distal tip of RIC-8::TagRFP ciliary signal (~3.5 μm) and measuring fluorescence intensities along the line using the plot profile tool. Fluorescence intensities within each cilium were normalized to the maximum intensity value for that cilium and expressed as percent of the maximum intensity. The mean percent of the maximum intensity was plotted at each ciliary position (n>20 animals).

*ODR-3::TagRFP fluorescence intensity (AWC)*: The *z* slices that encompassed the AWC cilium in its entirety (lateral view) were rendered into maximum-intensity projections in Fiji/Image J. Fluorescence intensity was quantified by outlining the AWC cilium using the freehand selection tool and measuring the mean fluorescence intensity. ODR-3::TagRFP fluorescence intensity for each condition in Fig 6D was normalized because this dataset was acquired on a different microscope from that in Fig 5. The normalization was performed for each condition in Fig 6D by subtracting the minimum value from the value of each datapoint and dividing by the range of values for that condition.

For quantification of fluorescence intensities in the distal dendrite encompassing the PCMC, a region of the dendrite extending 2 μm proximally from the cilia base was outlined using freehand selection tool, and the mean fluorescence intensity was recorded.

## Statistical analyses

Prism 9 software (GraphPad, San Diego, CA) was used to perform all statistical analyses and generate plots.

## Supporting information

**S1 Fig. Mutant RIC-8 proteins localize to cilia and do not disrupt cilia morphology.** (A–B) Representative images (A) and quantification (B) of localization patterns for the indicated RIC-8::TagRFP constructs in cilia of phasmid neurons. Anterior is at left; d–dendrite; purple arrowheads mark cilia base; scale bar: 5 μm. (C) Quantification of AWA cilium perimeter in adult animals of the indicated genotypes. Data for wild type are repeated from Fig 3C. In all scatter plots, means ± SD are indicated by horizontal and vertical black bars, respectively. The p-values were calculated using Kruskal-Wallis with Dunn's multiple comparisons test.
(TIF)

**S2 Fig. Internally tagged ODR-3 rescues *odr-3*-dependent cilia defects.** Representative images of ODR-3::TagRFP localization in AWC cilia of wild-type and *odr-3(n1605)* mutant adults. Numbers in top right corners indicate percentage of animals exhibiting the depicted phenotype (n>30/genotype). Anterior is at left; purple arrowheads mark cilia base; scale bar: 5 μm.
(TIF)

**S3 Fig. *ags-3* function is not required for chemotaxis toward benzaldehyde.** Chemotaxis responses of wild-type, *odr-3(n1605)*, and *ags-3(ok1169)* adult hermaphrodites to benzaldehyde diluted in ethanol (1:200). Dots–CI from single assays of approximately 200 animals each. Means ± SEM are indicated by horizontal and vertical black bars, respectively. *** indicates different from wild type at p < 0.001 (Brown-Forsythe and Welch ANOVA with Dunnett's T3 multiple comparisons test).
(TIF)

**S1 Table. List of *C. elegans* strains used in this work.**
(DOCX)

**S2 Table. List of plasmids used in this work.**
(DOCX)

**S3 Table. Numerical data used in graphs.**
(XLSX)

## Acknowledgments

We are grateful to Michael O'Donnell and members of the Nechipurenko lab for critical comments on the manuscript and Eric Peet for technical assistance and strain maintenance. We also thank Piali Sengupta, in whose lab these experiments were initiated, for support during the early stages of this project as well as comments and advice on the manuscript. Some strains were provided by the CGC, which is funded by NIH Office of Research Infrastructure Programs (P40 OD010440).

## Author Contributions

**Conceptualization:** Inna Nechipurenko.

**Formal analysis:** Christina M. Campagna.

**Funding acquisition:** Inna Nechipurenko.

**Investigation:** Christina M. Campagna, Hayley McMahon.

**Project administration:** Inna Nechipurenko.

**Resources:** Inna Nechipurenko.

**Supervision:** Inna Nechipurenko.

**Visualization:** Christina M. Campagna, Hayley McMahon, Inna Nechipurenko.

**Writing – original draft:** Christina M. Campagna, Inna Nechipurenko.

**Writing – review & editing:** Christina M. Campagna, Inna Nechipurenko.

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
