## [Decision Letter · Decision Letter 0]

22 Sep 2023

Dear Dr Nechipurenko,

Thank you very much for submitting your Research Article entitled 'The G protein alpha Chaperone and Guanine-Nucleotide Exchange Factor RIC-8 Regulates Cilia Morphogenesis in Caenorhabditis elegans Sensory Neurons' to PLOS Genetics.

The manuscript was fully evaluated at the editorial level and by independent peer reviewers. The reviewers appreciated the attention to an important topic but identified some minor concerns that we ask you address in a revised manuscript.  The concerns are minor and do not require for experimentation.  The reviewers and I felt it was well-written and provided important new insights.  Please address the few comments/concerns raised by the revieweres.

We therefore ask you to modify the manuscript according to the review recommendations. Your revisions should address the specific points made by each reviewer.

Yours sincerely,

Susan K. Dutcher

Academic Editor

PLOS Genetics

Gregory P. Copenhaver

Editor-in-Chief

PLOS Genetics

Reviewer's Responses to Questions

**Comments to the Authors:**

Reviewer #1: The manuscript by Campagna et al examines the role of a G-protein and its regulators in ciliary morphogenesis and ciliary function in C. elegans. Authors identify the GEF RIC-8 and activator of G-protein signaling AGS-3 as regulators of the ODR-3 G-protein, but via different mechanisms. Using a beautiful combination of genetics, in vivo imaging, structure-function studies, and behavioral analysis, authors show that both RIC-8 and AGS-3 are required for morphogenesis of the AWC wing cilia, but ASG-3 is not required for chemotaxis to benzaldehyde (ric-8 mutants are severely uncoordinated, precluding behavioral analysis). This result is exciting, because authors have genetically uncoupled AWC ciliary morphogenesis from AWC ciliary function, opening up many interesting questions and future experiments. This work is appropriate for the geneticist, cell biologist, and neuroscientist, and is appropriate for the readership of PLoS Genetics.

I have a few questions/comments that do not require additional experimentation.

(1) RIC-8 localization: Authors convincingly show that RIC-8 co-localizes with Inversin/NPHP-2S to the proximal region of rod cilia but not wing cilia. Do wing cilia possess an inversin compartment? What do authors think RIC-8 is doing in the inversin compartment of rod cilia? This would be an appropriate discussion point.

(2) RIC-8 and ODR-3 BIFC interaction: Authors use this clever strategy (with heat-shock RIC-8 to avoid developmental contributions) to show that RIC-8 and ODR-3 physically associate at specific regions of the AWC neuron (proximal region, ciliary base, NOT the entire wing). What is authors’ interpretation? This would also be an appropriate discussion point.

(3) Authors show that RIC-8 but not AGS-3 regulates ODR-3 levels and propose that RIC-8 is a chaperone for ODR-3. The data supports this hypothesis/model. ODR-3 levels/localization are not changed in the ags-3 mutant, however this ODR-3 overexpression rescues ags-3 AWC morphogenesis defects. This result complicates authors’ interpretation: “This result confirmed that cilia defects in ags-3 mutant are not due to reduced levels or mislocalization of ODR-3.” Would authors please comment on this?

(4) ODR-3 constitutively active overexpression phenotype (lines 361-363): “Interestingly, AWC distal dendrites…exhibited marked membranous extension at the ciliary base.” I agree but would be interested in hearing what authors think. It seems that this observation may relate to questions 1 and 2, which would be an appropriate discussion point.

(5) If the wings are not required for AWC chemosensation, what is their purpose? I assumed, wrongly, that the wings were to increase surface area for odorants and other small molecules.

Reviewer #2: In the manuscript by Campagna et al titled “The G protein alpha Chaperone and Guanine-Nucleotide Exchange Factor RIC-8 Regulates Cilia Morphogenesis in Caenorhabditis elegans Sensory Neurons” the authors describe a new functional role for a guanine exchange factor (GEF) and Gα chaperone, RIC-8, in the morphogenesis of wing cilia in C. elegans sensory neurons. RIC-8 localizes to cilia and LOF mutations of RIC-8 lead to altered cilia morphology. RIC-8 influences cilia morphology through its interactions with Gα and by modulating cGMP signaling. The paper is well written and organized, the data are convincing and provide an important fundamental insight. Understanding how cilia signaling and morphology are intertwined is critical to learning about their functions and roles in multiple systems including humans. I recommend this paper for publication.

Reviewer #3: This paper provides mechanistic insight of how a Galpha gene (odr-3), associated with GPCR signaling, operates in C. elegans sensory cilia to control cilia morphology and function. The authors focus on a quite well studied Galpha interactor, RIC8, which is known to serve GEF and chaperone functions. However, potential roles in relation to primary cilia have not been investigated. The authors show that RIC-8 localises within cilia, is required for expanding the membrane of a subset of cilia with complex morphologies, functions in the same pathway as odr-3 (based on double mutant analyses, and BiFC approaches), and positively regulates ODR-3 ciliary levels. Transgenic expression of ric-8 C-terminal truncation constructs in lof ric-8 mutant backgrounds shows that RIC-8’s cilia morphogenesis role depends on its chaperone-associated sequence, and that GEF activity alone is not sufficient for this function. The study then goes on to investigate AGS3, which is known to be involved in non-canonical G-protein signaling (AGS3 binding to Galpha subunits in the cytosol facilitates RIC8-mediated activation of the G-protein). The authors find that ags-3 is also required for cilia morphogenesis, with double mutant analyses indicating that ags-3 functions in the same pathway as ric-8. However, unlike ric-8, ags-3 does not regulate the ciliary levels of ODR-3, indicating that AGS-3 is not required for RIC-8’s chaperone function. Interestingly, despite abnormal morphology, the function of the examined cilia appear normal in ags-3 mutants, which suggests that non-canonical G-protein signaling is needed for cilium structure but not function. The paper finishes by showing that disruption of Gbeta/gamma signaling (by overexpressing the carboxy terminus of GRK-2 that blocks Gbeta/gamma binding with effectors) does not affect cilia morphology, thus supporting the conclusion that Galpha, rather than beta/gamma, signaling regulate cilium structure. From all the data, the authors present a comprehensive model of separable canonical and non-canonical Galpha signaling, involving distinct GEFs (GPCR – canonical; RIC-8 – non canonical) and binding factors (eg. AGS-3), that regulate the structure and function of a subset of C. elgans sensory cilia.

Overall, this is an impressive study, with conclusions that are generally well supported by the data. Almost all observations are quantified, and well controlled. The paper is very well written and presented. The nature of the topic will be of great interest to cilia biologists, as well as researchers with interests in G protein signaling.

I only have minor comments.

1. Line 112/113: Amphid neurons are not quite the majority of ciliated neurons in the hermaphrodite.

2. Line 155: the extent of the rescue is only partial. Thus, can the authors truly discount a cell non-autonomous role for ric-8 in addition to the cell autonomous role that is outlined ?

3. Fig. 4 C-E: do the authors see any BiFC signal in the absence of a heat shock? I would have thought that is an important control to show?

4. Line 296: Although RIC-8::GFP is still localised in the AWC cilia of odr-3 mutants, the signal distribution, at least as presented in Fig 5C, looks different to the control (mutant seems to show accumulation of GFP at the cilia endings). This should be commented on.

5. The study looks at benzaldehyde chemoattraction in ags-3 and odr-3 mutants; why were ric-8 mutants not assessed (is this due to a locomotion defect in these worms ?).

6. Line 384-393: it is very interesting that significant cilium structure defects in worms do not necessarily have to mean cilium function defects. It would be useful to the reader to know if such instances have been observed in mammals, given that many mammalian studies measure things like cilium length, with the implication being that shorter or longer cilia likely have cilium function defects. Perhaps this is not necessarily always so as is being shown here in worms. Maybe there is a tolerance of cilium length change that does not affect at least some cilia functions. I realise this is a significant topic to address but perhaps a couple of extra lines of thought on this matter is warranted in the discussion.

7. Figure 3A: indicate on the 1-566 schematic the positions of the GEF and the chaperone sequences.

8. Figure 3C: why no FL rescue data for the md1909 mutant?

9. Figure 4D: Interesting that the strongest BIFC signal is at the BB/PCMC region. Suggests that RIC-8 is preferentially binding ODR-3 outside the cilium, perhaps upstream of ODR-3s ciliary targeting? Can the authors comment on this?

**Have all data underlying the figures and results presented in the manuscript been provided?**

Reviewer #1: Yes

Reviewer #2: Yes

Reviewer #3: Yes

PLOS authors have the option to publish the peer review history of their article (what does this mean?). If published, this will include your full peer review and any attached files.

Reviewer #1: No

Reviewer #2: No

Reviewer #3: No

---

## [Editor Report · Decision Letter 1]

12 Oct 2023

Dear Dr Nechipurenko,

We are pleased to inform you that your manuscript entitled "The G protein alpha Chaperone and Guanine-Nucleotide Exchange Factor RIC-8 Regulates Cilia Morphogenesis in Caenorhabditis elegans Sensory Neurons" has been editorially accepted for publication in PLOS Genetics. Congratulations!

Yours sincerely,

Susan K. Dutcher

Academic Editor

PLOS Genetics

Gregory P. Copenhaver

Editor-in-Chief

PLOS Genetics

Comments from the reviewers (if applicable):

**Data Deposition**

http://datadryad.org/submit?journalID=pgenetics&manu=PGENETICS-D-23-00968R1

**Press Queries**

---

## [Editor Report · Acceptance letter]

26 Oct 2023

PGENETICS-D-23-00968R1 

The G protein alpha Chaperone and Guanine-Nucleotide Exchange Factor RIC-8 Regulates Cilia Morphogenesis in Caenorhabditis elegans Sensory Neurons 

Dear Dr Nechipurenko, 

We are pleased to inform you that your manuscript entitled "The G protein alpha Chaperone and Guanine-Nucleotide Exchange Factor RIC-8 Regulates Cilia Morphogenesis in Caenorhabditis elegans Sensory Neurons" has been formally accepted for publication in PLOS Genetics! Your manuscript is now with our production department and you will be notified of the publication date in due course.

With kind regards,

Anita Estes

PLOS Genetics

On behalf of:
